# The Interplay between Anticholinergic Burden and Anemia in Relation to 1-Year Mortality among Older Patients Discharged from Acute Care Hospitals

**DOI:** 10.3390/jcm10204650

**Published:** 2021-10-11

**Authors:** Andrea Corsonello, Luca Soraci, Francesco Corica, Valeria Lago, Clementina Misuraca, Graziano Onder, Stefano Volpato, Carmelinda Ruggiero, Antonio Cherubini, Fabrizia Lattanzio

**Affiliations:** 1Geriatric Medicine, IRCCS INRCA, 87100 Cosenza, Italy; a.corsonello@inrca.it; 2Unit of Geriatric Pharmacoepidemiology and Biostatistics, IRCCS INRCA, 87100 Cosenza, Italy; 3Department of Clinical and Experimental Medicine, University of Messina, 98124 Messina, Italy; coricaf@unime.it (F.C.); valeria.lago2@gmail.com (V.L.); 4Respiratory Unit, IRCCS INRCA, 23880 Casatenovo, Italy; c.misuraca@inrca.it; 5Department of Cardiovascular, Endocrine-Metabolic Diseases and Aging, Istituto Superiore di Sanitá, 00100 Rome, Italy; graziano.onder@iss.it; 6Center for Clinical Epidemiology, Department of Medical Sciences, School of Medicine, University of Ferrara, 44122 Ferrara, Italy; stefano.volpato@unife.it; 7Geriatric and Orthogeriatric Units, Geriatric and Gerontology Section, Department of Medicine and Surgery, University of Perugia, 06156 Perugia, Italy; carmelinda.ruggiero@unipg.it; 8Geriatria, Accettazione Geriatrica e Centro di Ricerca per l’invecchiamento, IRCCS INRCA, 60124 Ancona, Italy; a.cherubini@inrca.it; 9Scientific Direction, IRCCS INRCA, 60124 Ancona, Italy; f.lattanzio@inrca.it

**Keywords:** anticholinergic burden, anemia, hospital, older patients

## Abstract

Anticholinergic burden (ACB) and anemia were found associated with an increased risk of death among older patients. Additionally, anticholinergic medications may contribute to the development of anemia. Therefore, we aimed at investigating the prognostic interplay of ACB and anemia among older patients discharged from hospital. Our series consisted of 783 patients enrolled in a multicenter observational study. The outcome of the study was 1 year mortality. ACB was assessed by an Anticholinergic Cognitive Burden score. Anemia was defined as hemoglobin < 13 g/dL in men and <12 g/dL in women. The association between study variables and mortality was investigated by Cox regression analysis. After adjusting for several potential confounders, ACB score = 2 or more was significantly associated with the outcome in anemic patients (HR = 1.93, 95%CI = 1.13–3.40), but not non anemic patients (HR = 1.51, 95%CI = 0.65–3.48). An additive prognostic interaction between ACB and anemia was observed (*p* = 0.02). Anemia may represent a relevant effect modifier in the association between ACB and mortality.

## 1. Introduction

Anticholinergic medications are commonly used among older patients, despite the fact they are known to cause relevant side effects, including cognitive impairment and delirium, functional decline, disability and falls [1,2,3,4,5]. Older patients are particularly vulnerable to adverse reactions to anticholinergic drugs due to age-related changes in pharmacokinetics and pharmacodynamics [6], deficit of cholinergic transmission [7], comorbidity, polypharmacy, use of potentially inappropriate medications, and drug interactions [8,9]. Despite these potential risks, selected medications with some degree of anticholinergic activity, such as furosemide, digoxin, laxatives, and ranitidine are frequently taken in up to 37% of older people [10]. Even if each of the above example medications have weak anticholinergic properties, their co-administration frequently results in cumulative effects that are frequently unrecognized or misattributed as geriatric syndromes, frailty or simply changes due to ageing [11].

During the last decade, anticholinergic medications were repeatedly reported to be associated with reduced survival in several different populations, including community-dwelling individuals [1], nursing home residents [12], older hospitalized patients [13], and general older population [14]. More recently, several prognostic interactions involving anticholinergic burden were observed with risk factors relevant to the older population, including dependency in basic activities of daily living (BADL) [15], depression [16], physical [17] and cognitive impairment [18]. These findings are relevant from clinical point of view because they may help to identify patients carrying high risk of mortality in relation to cumulative exposure to anticholinergic medications who are likely to benefit of anticholinergic deprescribing.

Anemia is another relevant predictor of prognosis among older patients [19,20], and especially among hospitalized ones [21,22]. Low serum hemoglobin has shown to be a risk factor for 1 year mortality and increased hospitalization rates [23,24]. Additionally, anemia was previously shown to affect both cognitive [25,26] and physical performance [27,28] over time; indeed, persistence of low serum hemoglobin concentrations may induce a state of chronic brain hypoxia and reduce aerobic capacity, which may affect cognitive reserve and increase the risk of cognitive impairment and dementia [25]; moreover, anemia was also associated with a decline in physical performance [28,29] and sarcopenia [30]. Additionally, both cognitive and functional impairment were shown to increase the risk of 1 year mortality among older hospitalized patients with high ACB [15,18]. As a consequence, anemia may potentially mediate the negative effects of anticholinergic medications on cognitive, physical performance, and survival.

On the other hand, anticholinergic medications may contribute to the development of anemia by several different mechanisms, mainly represented by inhibition of iron absorption in the stomach and disruption of transferrin signaling [31,32]. Besides their effects on iron metabolism and transport, recent evidence suggests that nonselective anticholinergic medications may exert some detrimental effects on red blood cell (RBCs) turnover mainly by nonneuronal acetylcholine (Ach)-mediated modulation of hemorheological and oxygen-carrying properties of human erythrocytes [33] through M1 muscarinic receptors on RBCs [34] and bone marrow early erythroid progenitors [35]. However, despite biological plausibility, the potential prognostic interaction between anticholinergic burden and anemia has not been studied until now.

Therefore, the aim of this study was to investigate the prognostic interplay between anticholinergic burden and anemia in relationship with 1 year mortality. This study may help discover whether anticholinergic burden acts synergically with anemia and whether levels of circulating hemoglobin modulate the effect of anticholinergic burden on survival of older patients.

## 2. Materials and Methods

This present study was carried out using data from the CRiteria to assess Inappropriate Medication use among Elderly complex patients (CRIME) project, a multicenter prospective observational study involving seven geriatric and internal medicine acute wards in Italy. Methodology of CRIME project was described elsewhere [36]. Given that the CRIME study aimed at enrolling a real-world population of older in-patients, all patients aged 65 or older consecutively admitted to participating wards between June 2010 and May 2011 were asked to participate. The only exclusion criteria were being aged < 65 years and unwillingness to participate in the study. All study participants were asked to sign a written informed consent and were assessed within the first 24 hours from hospital admission and followed until discharge. Collected information included demographic, socioeconomic, and clinical characteristics, as well as detailed data about drug treatment and comprehensive geriatric assessment (CGA). Medications were coded according to the Anatomical Therapeutic and Chemical (ATC) classification [37]. All the drugs taken by the patients were carefully recorded before admission, during hospital stay and at discharge. Complete data about medications were also collected at 3-month follow-up visit. After discharge, patients were reassessed at 3, 6, and 12 months. The study was conducted in accordance with the Declaration of Helsinki, and the protocol was approved by the Ethics Committee of the Catholic University of Rome (Project identification code: P/582/CE/2009).

Overall, 1123 patients were enrolled in the present study. Patients with incomplete baseline data (*n* = 3) and those who died during hospitalization (*n* = 39) were excluded from the present analysis. Patients with incomplete follow-up data (*n* = 298) were also excluded, leaving a final sample of 783 patients to be included in the analysis. 

Patients excluded from the study were older (82.7 ± 7.3 vs. 80.9 ± 7.4, *p <* 0.001), more frequently females (60.8 % vs. 53.8 %, *p*= 0.034) and with lower number of medications (6.3 ± 3.5 vs. 7.5 ± 2.8, *p* < 0.001) compared to those included in the study. Additionally, they were also characterized by higher rates of cognitive impairment (68.5 % vs. 40.1%, *p* < 0.001), depression (48.1 % vs. 25.5%, *p* = 0.003), and BADL disability (51.6% vs. 32.8%, *p* < 0.001). 

### 2.1. Outcome

The outcome of the present study was 1 year mortality. Data on living status during the follow-up were obtained by interviewing the patients and/or their formal and/or informal caregivers. For patients who died during the follow-up period, the date and place of death were retrieved by relatives or caregivers. The municipal registers were consulted when neither patients nor relatives or caregivers could be contacted. 

### 2.2. Exposure Variables

Cumulative exposure to anticholinergic medications was assessed by the anticholinergic cognitive burden (ACB) score at discharge [38]. ACB score was chosen because of the availability of external validation and the greater accuracy in the assessment of central anticholinergic burden in comparison with other tools [39]. The main exposure variable was calculated as follows: ACB score at discharge, (1) low (ACB = 0, no ACB medications), (2) medium (ACB = 1), and (3) high burden (ACB = 2 or more). Anemia was defined by using WHO definition based on serum hemoglobin levels at discharge lower than 12 g/dL for females and 13 g/dL for males [40]. To investigate the impact of anemia on the relationship between ACB and prognosis, the ACB score at discharge was stratified by the presence or absence of anemia. 

### 2.3. Covariates

Age, sex, number of diagnoses, history of falls, and number of medications prescribed at discharge were considered as potential confounders in the analysis. CGA data were collected at the time of discharge. Patients with age- and education-adjusted Mini-Mental State Examination score of <24 were considered as cognitively impaired [41]. Geriatric Depression Scale score > 5 was used to identify patients with depression [42]. Dependency in at least 1 BADL was also considered as a potential confounder [43]. Selected diagnoses known to affect prognosis in older populations, including hypertension, heart failure, diabetes mellitus, atrial fibrillation, coronary artery disease (CAD), stroke, peripheral arterial disease (PAD), chronic obstructive pulmonary disease (COPD), chronic kidney disease (CKD), and cancer were also included in the analysis. Given the availability of complete data about medications at 3 months, ACB score at the 3-month follow-up visit was also considered as a potential confounder in order to explore the potential impact of changes in the exposure to anticholinergic medications over time. 

### 2.4. Analytic Approach

First, we analyzed the characteristics of patients according to ACB score at discharge among patients with or without anemia. The ꭕ^2^ test was used for categorical variables and one-way analysis of variance (ANOVA) for continuous ones. The association between exposure variables and the outcome was explored by Kaplan-Meier curves with log-rank test. Three different Cox proportional hazard model were used to estimate the HR and 95%CI for the effect of anemia and ACB score on 1 year mortality. The baseline model A was adjusted for age and sex; the multivariable model B was adjusted for all the variables associated with mortality in the preliminary analysis (age, sex, cognitive impairment, depression, history of falls, BADL disability, number of diagnoses, and number of medications); and model C including all variables from model B but specific diagnoses (hypertension, atrial fibrillation, heart failure, diabetes, CKD, PAD, CAD, COPD, cerebrovascular disease, and cancer) instead of number of comorbidities. Model C was also repeated after adjusting for ACB score at the 3-month follow-up or hemoglobin values. We used multivariable models to select predictors of mortality. To account for the impact of the severity of anemia on the observed associations, the analysis was also repeated using different cut-offs of hemoglobin [44]: mild anemia was defined as having hemoglobin of 11.0–12.9 g/dL for men or 11.0–11.9 g/dL for women; moderate-severe anemia as having hemoglobin values < 11.0 g/dL in both men and women. The interaction term ACB score at discharge*anemia was then formally investigated in Cox regression analysis; separate analyses were conducted among men and women to account for the impact of sex on the interaction term. Attrition bias was investigated by age- and sex-adjusted logistic regression analysis of ACB exposure to loss at the follow-up. 

Statistical analysis was carried out using R version 4.0 (R Foundation for Statistical Computing, Vienna, Austria, https://www.r-project.org/ (accessed on 8 October 2021)).

## 3. Results

Overall, anemia was diagnosed in 420 out of 783 patients (53.6%). The average ACB score at discharge was similar among patients with anemia and without anemia (median (IQR): 1 (0–2) vs. 1 (0–2), *p* = 0.19). ACB score categories (0, 1, 2 or more) were observed in 118 (28.1%), 154 (36.7%), and 148 (35.2%) patients with anemia, and in 130 (35.8%), 124 (34.2%), and 109 (30.0%) patients without anemia (*p = 0.06*). Among patients without anemia, those with ACB score =2 or more at discharge were older and more frequently affected by heart failure, atrial fibrillation, CAD, PAD, COPD, and cancer compared to patients with ACB = 0. BADL dependency, overall comorbidity and number of prescribed medications were also higher among patients with ACB score = 2 or more (Table 1). 

Among patients with anemia, those with ACB score = 2 or more had a greater prevalence of hypertension, heart failure, atrial fibrillation, and chronic kidney disease (CKD) and were characterized by a higher number of prescribed medications, a higher overall comorbidity, and a greater prevalence of cognitive impairment compared to those with ACB score = 0 (Table 1).

ACB medications prescribed at discharge among patients with or without anemia are reported in Table 2. 

ACB score showed a clear dose-response association with mortality in the whole study population (Table 3). ACB score at the 3-month follow-up was similar to that measured at discharge (no anemia: 1 (0–2); anemia: 1 (0–2), *p* = 0.25).

The graded increase in mortality in relation to ACB score at discharge was more evident among anemic compared to non-anemic patients (Figure 1). 

The association between ACB score at discharge and mortality in patients with anemia was confirmed after adjusting for potential confounders (Table 4). Furthermore, age (Hazard Ratio (HR) = 1.05, 95% Confidence Interval (CI) = 1.02–1.08), male sex (HR = 1.88, 95%CI = 1.30–2.74), and BADL dependency (HR = 3.15, 95%CI = 2.10–4.76) qualified as predictors of mortality in patients with anemia in model B; additional predictors of mortality in model C were CAD (HR = 1.74, 95%CI = 1.17–2.60), cancer (HR = 2.68, 95%CI = 1.78–4.03), and diabetes (HR = 1.52, 95%CI = 1.01–2.29). 

The association between ACB score of 2 or more and mortality in patients with anemia was also confirmed after adjusting model C for hemoglobin values at discharge (HR = 2.82, 95%CI = 1.29–6.12). Finally, after adjusting for ACB score at the 3 month follow-up, the association with mortality remained unchanged in patients with anemia and ACB score of 2 or more (HR = 2.91, 95%CI:1.34–5-68). Investigation of the prognostic impact of anemia severity on the observed associations showed that risk of mortality related to ACB score = 2 or more was similar among patients with mild anemia (HR = 1.73, 95%CI = 1.11–2.70) and those with moderate-severe anemia (HR = 1.88, 95%CI = 1.15–2.81). 

The ACB score was not significantly associated with survival among patients without anemia in adjusted analyses (Table 4). Predictors of mortality among patients without anemia were age (HR = 1.08, 95%CI = 1.03–1.13), and BADL dependency (HR = 4.46, 95%CI = 2.28–8.73) in model B, and age (HR = 1.07, 95%CI = 1.02–1.13), BADL dependency (HR = 5.76, 95%CI = 2.76–12.04), atrial fibrillation (HR = 2.45, 95%CI = 1.14–5.24), and cancer (HR = 4.61, 95%CI = 2.09–10.18) in model C. 

When we repeated the analysis using ACB score at discharge as a continuous variable instead of categorical one in the fully adjusted model C, the association with mortality was confirmed among patients with anemia (HR = 1.32, 95%CI = 1.12–1.55) and not among patients without anemia (HR = 1.20, 95%CI = 0.92–1.57). 

A mild but significant additive interaction between the ACB score at discharge and anemia was observed in the whole study population (*p* = 0.02). After stratifying Cox regression models by sex, the interaction between ACB score and anemia resulted to be strong and multiplicative among men (*p* < 0.001), and mild and additive among women (*p* = 0.03). The ACB score was not significantly associated with dropout rate, either in patients without anemia (ACB score = 1, odds ratio (OR) = 0.91; 95%CI = 0.83–1.00; ACB score = 2 or more, OR = 0.9; 95%CI = 0.88–1.06) or patients with anemia (ACB score = 1, OR = 0.98; 95%CI = 0.90–1.07; ACB score = 2 or more, OR = 1.05; 95%CI = 0.96–1.14). 

## 4. Discussion

Findings from the present study show that anticholinergic burden may be associated with reduced survival among older patients with anemia discharged from acute care wards in participating hospitals. Thus, anemia may have an additive effect on mortality risk in patients with ACB score = 2 or more, with a greater impact among men compared with women. The slightly high prevalence of ACB score = 2 or more among anemic patients may have also contributed to the observed findings. 

Anticholinergic medications share several central and peripheral adverse effects [45]. Mechanisms which may potentially account for ACB-associated adverse effects include cardiovascular (e.g., arrhythmias, syncope, ischemia) and neurologic (eg, hallucinations, confusion, seizure) side effects [46]. Moreover, age-related changes in pharmacokinetics and pharmacodynamics, as well as increased permeability of the blood-brain barrier and age-related acetylcholine depletion may favor the occurrence of adverse effects from anticholinergic medications among older patients [6,47]. Non-neuronal cholinergic system is disseminated on immunocompetent cells and the stimulation of nicotinic receptors may inhibit adaptive and innate immune reactions [48]. Consequently, anticholinergic drugs may harm by counteracting these immuno-modulatory actions leading to inflammation and increasing risk of death. 

It is worth noting that the prevalence of anemia was 53.6% in our study, which was higher than that formerly observed in the hospital setting (about 40%) [49], but very similar to the 53.5% prevalence recently observed among hospitalized patients aged 65 or older (48.3% among patients aged 65–80 years and 59.2% among patients aged > 80 years) [21]. 

Anticholinergic medications may potentially favor the development of anemia through several mechanisms: in general, anticholinergic substances may inhibit the absorption of iron at the gastric level [31], thus predisposing to hypochromic sideropenic anemia; additionally, several antipsychotics with strong anticholinergic properties may favor the development of anemia [50], via both direct and immune-mediated toxic actions upon the bone marrow or RBCs, and by decreasing body iron stores. Furthermore, the commonly used antiarrhythmic digoxin has anticholinergic properties and may predispose to development of anemia mainly disturbing transferrin signaling and iron storing [32].

On the other hand, anemia itself has negative prognostic outcomes in older people [20,21,22]. The findings of an increased risk of mortality among patients with either ACB score = 2 or more and anemia suggests that anemia may increase vulnerability to negative prognostic effects of anticholinergic drugs and/or anemia may mediate their negative prognostic effects. 

Potential reasons explaining the impact of anemia on the observed associations may be related to detrimental effects of nonselective anticholinergic medications on RBCs turnover. RBCs are in fact very effective scavengers of nonneuronal acetylcholine (ACh) escaping into the bloodstream [34]. Nonneuronal ACh is able to modulate the hemorheological and oxygen-carrying properties of human erythrocytes [33] mainly through muscarinic receptors of type M1 which have been found in high density on surface of RBCs [34] and bone marrow early erythroid progenitors [35]. Changes in the RBCs’ hemorheological properties may trigger changes in blood viscosity and modulate tissue oxygenation and the distribution of blood to the peripheral tissues [33,34]. It is thought that acetylcholine down-regulates the self-renewal of RBCs and bone marrow erythroid precursors, since pharmacologic inhibition or genetic suppression of cholinergic receptors, muscarinic 4 (CHRM4) has shown to improve RBCs production in both in vitro and in vivo studies [35]. ACh action is limited to the internal environment of erythrocytes by the activity of acetylcholinesterase (AChE), an enzyme involved in its breakdown, that is highly expressed on the RBC membrane, and contribute to maintaining the size and shape of RBCs [51]. AChE seems also to mediate erythroid differentiation, working in association with erythropoietin (EPO) with a feedback-loop mechanisms: on one hand, EPO induces the transcription of AChE genes, while AChE increases the responsiveness of erythroid cells to EPO [52]; the final effect of this interaction is the increase in RBCs production. However, AChE overexpression is a reliable marker of aging, inflammatory states, and several diseases, such as hypertension, glaucoma, dementia, and anemia [51,53]. Additionally, scopolamine (i.e., a nonselective anticholinergic medication) was found to increase AChE activity in several different experimental models [54,55], though the role of anticholinergic-induced AChE overexpression in the pathophysiology of anemia is still to be elucidated. 

Patients with anemia may have also high susceptibility to negative iatrogenic events. In fact, anemia is often associated with sarcopenia [56], which may in turn change the volume of distribution of several drugs, thus affecting pharmacokinetics and pharmacodynamics response to selected drugs and increasing the risk of iatrogenic adverse reactions. Moreover, both anemia and high anticholinergic burden are risk factors for cognitive impairment [57,58], which may in turn increase patient’s vulnerability to iatrogenic side effects of anticholinergic drugs mainly by decreasing individual autonomy and adherence to drug regimens [59]. Additionally, ACB score was found associated with BADL dependency [15] and depression [16], that may both increase mortality of older patients and were proved to be associated with anemia [60]. 

Awareness of the excessive mortality risk associated with the use of anticholinergic medications should lead physicians to limit their prescription, especially among older patients with anemia. The association between ACB score = 2 or more and mortality among anemic patients was mainly driven by cumulative use of drugs with low anticholinergic effect in our study. However, it is worth noting that a not negligible proportion of patients (39 out of 363 in non-anemic group and 30 out of 420 in anemic group) were prescribed medications with moderate-high anticholinergic activity. 

Thus, deprescribing of anticholinergic medications warrants further investigations. Meanwhile, hospitalization should always be considered a clue to identify anemia and to select drugs with no or less anticholinergic burden whenever possible (e.g., avoiding tricyclics, trazodone, or paroxetine). A slow and gradual withdrawal of anticholinergic medications should always be started when indications to their use are no longer present, especially among anemic patients. 

Limitations of our study deserve to be mentioned. Given the observational design, confounding by indication cannot be ruled out. Patients excluded from the analysis because of incomplete follow-up had greater overall comorbidity and number of medications, as well as higher prevalence of selected diagnoses, cognitive and functional impairment. However, attrition bias analysis showed that ACB score was not associated with drop-out rate. Additionally, our results identify variables that by themselves may influence the outcome, and we could not account for illness severity, duration and management of individual diagnoses, and life expectancy. Furthermore, we cannot rule out that measures of cumulative anticholinergic burden other than ACB may yield different findings. Similarly, anemia was defined on the basis of circulating hemoglobin, and lack of data on RBCs quantity and morphology, other laboratory parameters, as well as duration of anemia, did not allow us to explore the impact of different forms of anemia on study outcomes. The short duration of the follow-up, up to 12 months, limited the study of the association between ACB and prognosis. Our dataset did not allow to investigate competing risk related to readmissions and/or emergency department visits during follow-up. Similarly, our dataset did not allow us to investigate the prognostic weight of frailty or IADL, as BADL scale was the only available measure to investigate physical dependency. Additionally, ACB data during follow-up were only available at 3 months, which limits the exploration of longer exposure to anticholinergics. Furthermore, the small sample size may reduce the precision of estimates and does not allow to preform dose-response analysis after stratification by anemia. Thus, the finding of a not significant trends for association between ACB score and mortality among patients without anemia does not mean that ACB drugs can be considered safe in these patients. Finally, our results apply to a population of older patients discharged from acute care hospitals with a diagnosis of anemia and could not be generalized or applied to other settings. Thus, further research using larger population samples with extended follow-up periods, as well as confirmatory studies with other different measures of anticholinergic burden are needed. Nevertheless, the inclusion of a real-world population of hospitalized older patients, the detailed assessment of drugs taken by each individual patients, as well as the systematic use of CGA which allowed us to adjust the analysis for a wide set of potential confounders should be considered as relevant strengths.

## 5. Conclusions

ACB score at discharge is a relevant predictor of 1 year mortality among older patients discharged from acute care hospitals; anemia was found to modulate the relationship between ACB score and mortality. For this reason, hospital physicians should be aware that prescribing anticholinergic medications in such a vulnerable population may have a negative prognostic impact. Thus, hospitalization should be a clue to identify patients with anemia and to revise overall drug treatment to reduce ACB at discharge whenever possible. 

## Figures and Tables

**Figure 1 jcm-10-04650-f001:**
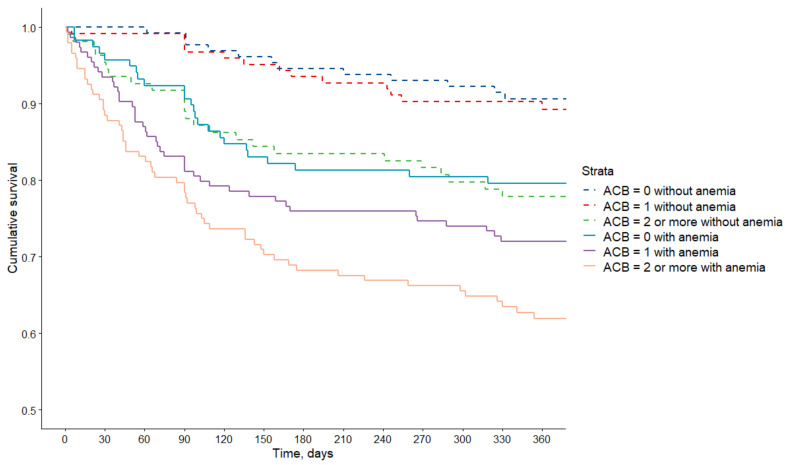
Kaplan Meier curves showing survival associated with ACB score and anemia.

**Table 1 jcm-10-04650-t001:** Demographic and clinical characteristics of patients stratified by anemia and ACB score at discharge.

		No Anemia (*n* = 363)				Anemia (*n* = 420)			
		ACB score at discharge				ACB score at discharge			
	All patients (*n* = 783)	0 (*n* = 130)	1 (*n* = 124)	2 or more (*n* = 109)	*p* value ^a^	0 (*n* = 118)	1 (*n* = 154)	2 or more (*n* = 148)	*p* value ^a^
Age, mean (± SD)	80.9 ± 7.4	76.9 ± 6.9	78.8 ± 7.3	80.7 ± 7.6	<0.001	82.5 ± 7.5	83.3 ± 6.6	82.6 ± 6.7	0.98
Male sex, *n* (%)	362 (46.2)	55 (42.3)	66 (53.2)	51 (46.8)	0.22	48 (40.7)	76 (49.3)	66 (44.6)	0.36
Cognitive impairment, *n* (%)	314 (40.1)	41 (31.5)	51 (41.1)	49 (45.0)	0.09	45 (38.1)	76 (49.3)	52 (35.1)	0.03
Depression, *n* (%)	200 (25.5)	31 (23.8)	29 (23.4)	31 (28.4)	0.62	24 (20.3)	43 (27.9)	42 (28.4)	0.26
History of falls, *n* (%)	198 (25.3)	25 (19.2)	27 (21.8)	33 (30.3)	0.12	28 (23.8)	50 (32.5)	35 (23.6)	0.15
BADL disability, *n* (%)	257 (32.8)	22 (16.9)	22 (17.7)	38 (34.9)	0.001	43 (36.4)	65 (42.2)	67 (45.3)	0.34
Number of diseases, mean (± SD)	5.2 ± 2.7	3.7 ± 2.0	4.7 ± 2.0	5.7 ± 2.7	<0.001	4.7 ± 2.7	5.8 ± 2.7	6.4 ± 2.8	<0.001
Hypertension, n (%)	595 (76.0)	106 (81.5)	103 (83.1)	82 (75.2)	0.29	68 (57.6)	117 (76.0)	119 (80.4)	<0.001
Heart failure, n (%)	223 (28.5)	6 (4.6)	30 (24.2)	42 (38.5)	<0.001	11 (9.3)	64 (41.6)	70 (47.3)	<0.001
Diabetes, n (%)	234 (29.9)	26 (20.0)	36 (29.0)	31 (28.4)	0.19	41 (34.7)	46 (29.9)	54 (36.5)	0.45
Atrial fibrillation, n (%)	144 (18.4)	5 (3.8)	14 (11.3)	22 (20.2)	<0.001	11 (9.3)	33 (21.4)	59 (39.9)	<0.001
CAD, n (%)	245 (31.3)	17 (13.1)	38 (30.6)	44 (40.4)	<0.001	29 (24.6)	55 (35.7)	62 (41.9)	0.01
Cerebrovascular disease, n (%)	157 (20.0)	24 (18.5)	25 (20.1)	23 (21.1)	0.87	22 (18.6)	32 (20.8)	31 (20.9)	0.88
PAD, n (%)	64 (8.2)	3 (2.3)	14 (11.3)	8 (7.3)	0.02	7 (5.9)	16 (10.4)	16 (10.8)	0.33
CKD, n (%)	409 (53.3)	42 (33.1)	56 (46.7)	52 (49.1)	0.02	56 (49.1)	101 (66.0)	102 (69.4)	0.002
COPD, n (%)	303 (38.7)	42 (32.3)	59 (47.6)	45 (41.3)	0.04	37 (31.3)	54 (35.1)	66 (44.6)	0.06
Cancer, n (%)	109 (13.9)	4 (3.1)	8 (6.4)	18 (16.5)	<0.001	25 (21.2)	28 (18.2)	26 (17.6)	0.73
Number of medications, mean (± SD)	7.5 ± 2.8	5.8 ± 2.4	8.0 ± 2.3	8.6 ± 2.7	<0.001	5.6 ± 2.6	7.7 ± 2.5	9.0 ± 2.6	<0.001

Notes: BADL = Basic Activity of Daily Living; CAD= Coronary artery disease; CKD = Chronic Kidney Disease; COPD = Chronic Obstructive Pulmonary Disease; PAD = Peripheral Arterial Disease; SD = Standard Deviation. (^a^) *p* values are from Chi-square or ANOVA 1-way test as appropriate.

**Table 2 jcm-10-04650-t002:** Anticholinergic Cognitive Burden (ACB) listed medications prescribed at discharge in the study population according to diagnosis of anemia.

	No Anemia (*n* = 363)	Anemia (*n* = 420)
ACB score 1	Furosemide 156 (43.0%) Prednisone 25 (6.9%) Metoprolol 23 (6.3%) Digoxin 14 (3.8%) Isosorbide 20 (5.5%) Codeine 17 (4.7%) Warfarin 9 (2.5%) Alprazolam 10 (2.8%) Atenolol 10 (2.7%) Trazodone 7 (1.9%) Ranitidine 6 (1.6%) Chlortalidone 2 (0.5%) Cetirizine 2 (0.5%) Haloperidol 2 (0.5%) Diazepam 3 (0.8%) Theophylline 2 (0.5%) Colchicine 1 (0.2%) Risperidone 1 (0.2%) Captopril 1 (0.2%) Aripiprazole 1 (0.2%)	Furosemide 243 (57.8%) Prednisone 41 (9.8%) Digoxin 44 (10.5%) Metoprolol 28 (6.7%) Isosorbide 20 (4.8%) Codeine 12 (2.9%) Warfarin 12 (2.9%) Trazodone 12 (2.9%) Atenolol 9 (2.1%) Alprazolam (1.9%) Ranitidine 7 (1.7%) Haloperidol 5 (1.2%) Fentanyl 4 (0.9%) Risperidone 2 (0.5%) Hydrocortisone 2 (0.5%) Chlortalidone 1 (0.2%) Cetirizine 1 (0.2%) Diazepam 1 (0.2%) Theophyilline 1 (0.2%) Aripiprazole 1 (0.2%)
ACB score 2	Carbamazepine 4 (1.1%) Oxcarbazepine 1 (0.2%)	Carbamazepine 4 (0.9%) Oxcarbazepine 1 (0.2%) Meperidine 1 (0.2%)
ACB score 3	Quetiapine 16 (9.6%) Paroxetine 6 (1.6%) Promazine 5 (1.4 %) Amitryptiline 3 (0.8%) Scopolamine 1 (0.2%) Clomipramine 1 (0.2%) Oxybutynin 1 (0.2%) Orphenadrine 1 (0.2%)	Quetiapine 16 (3.8%) Promazine 5 (1.2%) Paroxetine 3 (0.7%) Scopolamine 1 (0.2%)

**Table 3 jcm-10-04650-t003:** Cox regression analysis exploring dose-response relationship between ACB score and mortality in the whole study population.

ACB Score at Discharge	Mortality Rate ^a^	HR (95%CI) ^b^
0	36 (14.5%)	1.0
1	56 (20.1%)	1.29 (0.82–2.01)
2	38 (27.7%)	1.63 (0.98–2.71)
3	21 (31.3%)	1.96 (1.09–3.52)
4	14 (37.8%)	2.34 (1.21–4.53)
5	8 (53.3%)	5.49 (2.42–12.44)

Notes: ^a^. Data are number of cases (percentages). ^b^. Cox regression model was adjusted for age, sex, cognitive impairment, depression, BADL disability, history of falls, number of drugs and number of diseases.

**Table 4 jcm-10-04650-t004:** Cox proportional hazard models of the relationship between ACB score at discharge and 1 year mortality stratified by the presence of anemia.

	Mortality Rate (%)	Model A HR (95%CI)	Model B HR (95%CI)	Model C HR (95%CI)
ACB score at discharge				
No anemia (*n* = 363)				
0	12 (9.2)	1.0	1.0	1.0
1	13 (10.5)	0.96 (0.43–2.11)	0.96 (0.40.2.28)	1.02 (0.39–2.64)
2 or more	25 (22.9)	1.88 (0.93–3.81)	1.51 (0.65–3.48)	1.27 (0.49–3.25)
Anemia (*n* = 420)				
0	24 (20.3)	1.0	1.0	1.0
1	43 (27.9)	1.35 (0.82–2.23)	1.35 (0.79–2.29)	1.39 (0.77–2.50)
2 or more	56 (37.8)	2.20 (1.36–3.55)	1.96 (1.13–3.40)	1.97 (1.06–3.67)

Notes: ACB = Anticholinergic Cognitive Burden. CI = Confidence Interval; HR = Hazard Ratio. Model A, adjusted for age and sex; model B, adjusted for age, sex, cognitive impairment, depression, history of falls, BADL disability, number of diagnoses, and number of medications; and model C, adjusted for age, sex, cognitive impairment, depression, history of falls, BADL disability, number of medications, and specific diagnoses (hypertension, atrial fibrillation, heart failure, diabetes, CKD, PAD, CAD, COPD, cerebrovascular disease, cancer) instead of number of diagnoses.

## Data Availability

Data are available for CRIME study researcher at IRCCS INRCA (www.inrca.it (accessed on 8 October 2021)).

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
