# Peer review of "The Interplay between Anticholinergic Burden and Anemia in Relation to 1-Year Mortality among Older Patients Discharged from Acute Care Hospitals"

_jcm, 2021, doi:10.3390/jcm10204650_

Round 1
Reviewer 1 Report
This interesting study investigates the prognostic interplay between anticholinergic burden and anemia in relation to 1-year mortality. Studies have been well-conducted around the central effects of ACH burden; however, researchers haven’t explored other under-reported effects such as anaemia. The authors need to make a strong justification in the introduction and describe why this particular outcome is so important when the rest of the world is worried about cognitive and physical outcomes? In my opinion, despite this current evaluation of the manuscript, the information reported needs some clarification.
- Why had the authors chosen BADL instead of IADL/ADL?
- In the methods, it's unclear whether the pts had anaemia which is newly diagnosed or pre-existing (if so, for how many years)?
- What types of anaemia that the authors categorised here? Microcytic (iron-deficiency), haemolytic, etc. What evidence is currently supporting ACH effects and types of anaemia?
- ‘Anemia is a common complication of chronic kidney disease (CKD)’. I can see that the study included CKD as a variable. How did the authors adjust their model for this potential confounder?
- Please clarify – what was the relevance of adjusting model C for Hb values? Was the intention to classify mild, moderate and severely anaemic patients? Was there any cut-off?
- The findings are interesting. High anticholinergic burden itself is a proxy indicator for mortality (https://pubmed.ncbi.nlm.nih.gov/31832732/). I am not getting a clear picture of how anaemia can contribute to reduced survival? Please clarify
- Please be mindful of the abbreviations (AChE, not AchE).
- Line 26 (ACB already defined in line 23)
- Line number 29: <13 g/dL
- Line number 69: Delete ‘uses’
- Line number 74: aged
- Line number 74: Inclusion criteria is not clear. I suggest elaborating it before stating exclusion criteria.
- Line abstract 114: Abbreviation for comprehensive geriatric assessment had already been used earlier in methodology.
- Line number 117-118: Need to be consistent in using the abbreviations for BADL.
- Line number 118: I suggest using a recent reference. It’s an extremely old reference (1963).
- Line number 119: diabetes mellitus?
- Line number 147-148: Need to correct errors in using abbreviations. All these terms had already been abbreviated.
- Line number 151: Need to state value of higher in the bracket. For example, 2 or more
- Line number 211: Is it acute care wards or hospitals? This term was earlier not used anywhere in methodology except acute wards. Need to clarify.
- Line number 212-213: statistically significant only among patients with low haemoglobin levels? I couldn’t verify it from the study findings. In addition, it blurred whether it was a specific gender or both?
- Line number 242: Ach needs to define first before using an abbreviation. Also, the international acronym is ACh
- Line number 245: Abbreviation for red blood cells should have been used earlier in line number 241
- Line number 25: Replace red blood cells with ‘RBCs’
- Line number 258: RBC
- Line number 289: burden
- Line number 291: RBC’s quantity….
- Line number 292: Did not allow us……
- Line number 304: CGA
- Line number 413: Study title is in a capitalise format
- I also noticed that this is a continuation of the author's previous study https://pubmed.ncbi.nlm.nih.gov/29292589/; was there any difference noticed this time? I heeded the use of several statements from your previous study without paraphrasing. Please modify those bits.
Author Response
This interesting study investigates the prognostic interplay between anticholinergic burden and anemia in relation to 1-year mortality. Studies have been well-conducted around the central effects of ACH burden; however, researchers haven’t explored other under-reported effects such as anaemia. The authors need to make a strong justification in the introduction and describe why this particular outcome is so important when the rest of the world is worried about cognitive and physical outcomes?
AU] We would like to thank the Reviewer for this comment which allowed us to significantly improve our work. The introduction has been extensively revised according to the Reviewer’s suggestion. Description of the relationship between ACB and anemia has been also improved in the Introduction.
In my opinion, despite this current evaluation of the manuscript, the information reported needs some clarification.
- Why had the authors chosen BADL instead of IADL/ADL?
AU] We agree with the reviewer that IADL would be useful for a better definition of functional status. Unfortunately, our dataset did not include the assessment of IADL disability. We recognized this limitation on lines 357-359.
- In the methods, it's unclear whether the pts had anaemia which is newly diagnosed or pre-existing (if so, for how many years)?
AU] Even in this case we completely agree with the Reviewer. Unfortunately our dataset did not allow to investigate this issue. We added a limitation on lines 351-354
- What types of anaemia that the authors categorised here? Microcytic (iron-deficiency), haemolytic, etc. What evidence is currently supporting ACH effects and types of anaemia?
AU Unfortunately our dataset did not allow to investigate this issue. We added a limitation on lines 351-354
- ‘Anemia is a common complication of chronic kidney disease (CKD)’. I can see that the study included CKD as a variable. How did the authors adjust their model for this potential confounder?
AU] We improved description of covariates included in the adjusted model C (hypertension, heart failure, diabetes, CAD, PAD, COPD, cerebrovascular disease, cancer, CKD): see footnotes of table 4 and lines 156-163 in the methods section.
- Please clarify – what was the relevance of adjusting model C for Hb values? Was the intention to classify mild, moderate and severely anaemic patients? Was there any cut-off?
AU] We clarified the reason we adjusted model C and cut-off used for grading the severity of anemia at lines 165-169 of the methods and 235-238 of the results.
- The findings are interesting. High anticholinergic burden itself is a proxy indicator for mortality (https://pubmed.ncbi.nlm.nih.gov/31832732/). I am not getting a clear picture of how anaemia can contribute to reduced survival? Please clarify
AU] The Introduction has been extensively revised, and prognostic and clinical implication of anemia discussed.
- Please be mindful of the abbreviations (AChE, not AchE).
AU] We corrected as requested
- Line 26 (ACB already defined in line 23)
AU] We corrected as requested
- Line number 29: <13 g/dL
AU] We corrected as requested
- Line number 69: Delete ‘uses’
AU] We corrected as requested
- Line number 74: aged
AU] We corrected as requested
- Line number 74: Inclusion criteria is not clear. I suggest elaborating it before stating exclusion criteria.
AU] We clarified that the CRIME study aimed at enrolling a real-world population of older in-patients. For this reason, all patients aged 65 or more consecutively admitted to participating wards between June 2010 and May 2011 were asked to participate. The only exclusion criteria were being aged < 65 years and unwillingness to participate in the study (lines 95-99).
- Line abstract 114: Abbreviation for comprehensive geriatric assessment had already been used earlier in methodology.
AU] We corrected as requested
- Line number 117-118: Need to be consistent in using the abbreviations for BADL.
AU] We corrected and checked BADL abbreviation throughout the manuscript
- Line number 118: I suggest using a recent reference. It’s an extremely old reference (1963).
AU] We cited the original reference on JAMA. However, a more recent reference regarding BADL use among older hospitalized patients was added in the present version.
- Line number 119: diabetes mellitus?
AU] We corrected as requested
- Line number 147-148: Need to correct errors in using abbreviations. All these terms had already been abbreviated.
AU] We corrected as requested
- Line number 151: Need to state value of higher in the bracket. For example, 2 or more
AU] We corrected as requested throughout the manuscript
- Line number 211: Is it acute care wards or hospitals? This term was earlier not used anywhere in methodology except acute wards. Need to clarify.
AU] We clarified that we refer to acute care wards in participating hospitals (lines 259-260)
- Line number 212-213: statistically significant only among patients with low haemoglobin levels? I couldn’t verify it from the study findings. In addition, it blurred whether it was a specific gender or both?
AU] We clarified the sentence at lines 261-263 of the discussion based on study results and table 4. We included analysis of difference in the interaction term among men and women to account for the impact of sex on such interaction. See lines 171-172 in the methods part and lines 250-253 in the results.
- Line number 242: Ach needs to define first before using an abbreviation. Also, the international acronym is Ach
AU] We corrected as requested throughout the manuscript
- Line number 245: Abbreviation for red blood cells should have been used earlier in line number 241
AU] Red blood cell has been abbreviated at first citation and abbreviation used throughout the manuscript
- Line number 25: Replace red blood cells with ‘RBCs’
AU] Red blood cell has been abbreviated at first citation and abbreviation used throughout the manuscript
- Line number 258: RBC
AU] Red blood cell has been abbreviated at first citation and abbreviation used throughout the manuscript
- Line number 289: burden
AU] We corrected as requested
- Line number 291: RBC’s quantity….
AU] Red blood cell has been abbreviated at first citation and abbreviation used throughout the manuscript
- Line number 292: Did not allow us……
AU] We corrected as requested
- Line number 304: CGA
AU] We used abbreviation as requested
- Line number 413: Study title is in a capitalise format
AU] We used capitalisation to identify characters contributing to study acronym
- I also noticed that this is a continuation of the author's previous study https://pubmed.ncbi.nlm.nih.gov/29292589/; was there any difference noticed this time? I heeded the use of several statements from your previous study without paraphrasing. Please modify those bits.
AU] Introduction and discussion were extensively revised and overlapping with previous paper reduced. The study is consistently different compared to former one based on aim, sample selection, results and clinical implications.
Reviewer 2 Report
Comments to the Author
Thank you for the opportunity to review the manuscript entitled ‘The interplay between anticholinergic burden and anemia in relation to 1-year mortality among older patients discharged from acute care hospitals.’
General comment: The manuscript is well-written. The topic is very relevant to the older population as anticholinergic medications are still widely utilized among older adults, despite several reports of multiple safety issues.
Abstract
- Line 29 – ‘hemoglobin<13 g/del’ to ‘hemoglobin<13 g/dl’
Introduction
- On Line 44 – “Despite these potential risks, selected anticholinergic medications, such as furosemide, digoxin, laxatives and ranitidine are frequently taken in up to 37%....”. Authors emphasized more on medications with weak anticholinergic activity, i.e., ACB score of 1, rather it would have been better to focus more on medication with known/ strong anticholinergic activity (i.e., ACB score of 2 or more). I would prefer the term medications with anticholinergic activity when discussing those with minimal/weak anticholinergic activity, such as the medications mentioned in the above statement. Please can the authors indicate that those medications listed on Line 44 (i.e., furosemide, digoxin, laxatives, and ranitidine) are medications with weak anticholinergic effects. And emphasize the total anticholinergic burden could build up when multiple medications with weak anticholinergic activity are used concomitantly.
- On Line 50 and 51 – can the authors include the relevant reference (s) after mentioning each risk factor - BADL, depression, physical and cognitive impairment rather than putting at the end of the line. [ It is already done in the discussion section]
- On Line 59-60 – same as above – include relevant ref/s after each proposed mechanism of action. Also, for the readers to appreciate the clinical relevance – can the authors include more discussion on the biological plausibility between ACB exposure and anemia. The authors mentioned broad mechanisms but how could an anticholinergic medication causally be linked with anemia needs further clarification. Is there any data, how many percent of medications listed in the ACB score can be causally linked with anemia? This is very important for this paper. What about the other way – low hemoglobin level could potentially augment the anticholinergic induced adverse effects thereby mortality? Also, need a bit of statement on the link between ACB and mortality in older adults from previous work before concluding the intro part.
Materials and Methods
- On Line 123 – can the authors clarify why only exposure to anticholinergic medications at 3-month considered but not at 6-month follow up? – got it at the study limitation section. Can the authors include the reasons firsthand in the method section?
- Covariates - what about the impact of readmission or ED visits during the 1-year follow up after discharge – it is a common event among hospitalized older adults. Hospital readmission or ED visit events as a competing risk?
Results
- On Line 143-144 – there is statistical difference on ACB score categories between participants with and without anemia- what is the implication of this difference on the main outcomes between the two groups-e.g., ACB score 2 or more 35.2% anemic vs 30.0% without anemia. What is the implication of having more participant in the anemic group with higher ACB score than in those without anemia? Can this impact the observed difference in the outcome between groups?
- On Line 173: Figure 1 – I couldn’t differentiate the strata legend – can the authors use colored figure?
- On Line 176 – 180 and Line 194-197 authors presented predictors of mortality, however, there is no any specific analysis plan included in the methods part. To present predictors of an outcome, the authors need to describe their approach for variable selection …what is mentioned in the methods is that they run different Cox models with d/t level of covariate adjustment. It is also not clearly mentioned which model is used to define “predictors” or which model (A, B, C) is the primary analysis? Predictor analysis requires extensive list of variables.
Discussion
- Can the authors discuss the high prevalence of anemia in the study cohort – 53.6% is a bit high prevalence in older adults but not uncommon. Can they add some sentences about this high prevalence?
- Line 242 - include the abbreviation – “nonneuronal acetylcholine (Ach)”
- Excellent discussion on the biological plausibility of AC exposure and anemia – can the authors include a summary of other mechanisms included in the discussion but not reflected in the intro part.
- Can the authors expand the clinical implication of the findings on Line 274-79 (especially focusing on strong anticholinergics with ACB 2 or more score as this was the significant one from the analysis), including reducing cumulative anticholinergic burden via deprescribing interventions in older adults with anemia as well as future research gaps (for instance further research using large sample for extended periods, confirmation study with other measures of anticholinergic burden, e.g. DBI…).
- Line 239 – spelling error ‘measures of cumulative anticholinergic burde’.
Author Response
Thank you for the opportunity to review the manuscript entitled ‘The interplay between anticholinergic burden and anemia in relation to 1-year mortality among older patients discharged from acute care hospitals.’
General comment: The manuscript is well-written. The topic is very relevant to the older population as anticholinergic medications are still widely utilized among older adults, despite several reports of multiple safety issues.
Abstract
- Line 29 – ‘hemoglobin<13 g/del’ to ‘hemoglobin<13 g/dl’
AU] We corrected as requested.
Introduction
- On Line 44 – “Despite these potential risks, selected anticholinergic medications, such as furosemide, digoxin, laxatives and ranitidine are frequently taken in up to 37%....”. Authors emphasized more on medications with weak anticholinergic activity, i.e., ACB score of 1, rather it would have been better to focus more on medication with known/ strong anticholinergic activity (i.e., ACB score of 2 or more). I would prefer the term medications with anticholinergic activity when discussing those with minimal/weak anticholinergic activity, such as the medications mentioned in the above statement. Please can the authors indicate that those medications listed on Line 44 (i.e., furosemide, digoxin, laxatives, and ranitidine) are medications with weak anticholinergic effects. And emphasize the total anticholinergic burden could build up when multiple medications with weak anticholinergic activity are used concomitantly.
AU] We corrected as requested
- On Line 50 and 51 – can the authors include the relevant reference (s) after mentioning each risk factor - BADL, depression, physical and cognitive impairment rather than putting at the end of the line. [ It is already done in the discussion section]
AU] We corrected as requested
- On Line 59-60 – same as above – include relevant ref/s after each proposed mechanism of action.
AU] We corrected as requested
Also, for the readers to appreciate the clinical relevance – can the authors include more discussion on the biological plausibility between ACB exposure and anemia. The authors mentioned broad mechanisms but how could an anticholinergic medication causally be linked with anemia needs further clarification. Is there any data, how many percent of medications listed in the ACB score can be causally linked with anemia? This is very important for this paper. What about the other way – low hemoglobin level could potentially augment the anticholinergic induced adverse effects thereby mortality? Also, need a bit of statement on the link between ACB and mortality in older adults from previous work before concluding the intro part.
AU] We changed the Introduction in keeping with the Reviewer’s suggestion. In particular, we briefly introduced some details in regard to biological plausibility of our main hypothesis. Additionally, we also stated about the association between anticholinergic burden and mortality in former studies.
Materials and Methods
- On Line 123 – can the authors clarify why only exposure to anticholinergic medications at 3-month considered but not at 6-month follow up? – got it at the study limitation section. Can the authors include the reasons firsthand in the method section?
AU] We clarified that complete data about medications were also collected at 3-month follow-up visit. For this reason, ACB score at 3-month was also considered in the analysis (lines 147-150).
- Covariates - what about the impact of readmission or ED visits during the 1-year follow up after discharge – it is a common event among hospitalized older adults. Hospital readmission or ED visit events as a competing risk?
AU] Unfortunately, our dataset did not allow to investigate this issue. We recognized this limitation on lines 355-357
Results
- On Line 143-144 – there is statistical difference on ACB score categories between participants with and without anemia- what is the implication of this difference on the main outcomes between the two groups-e.g., ACB score 2 or more 35.2% anemic vs 30.0% without anemia. What is the implication of having more participant in the anemic group with higher ACB score than in those without anemia? Can this impact the observed difference in the outcome between groups?
AU] We described the potential impact of this different prevalence at lines 262-263 in the discussion.
- On Line 173: Figure 1 – I couldn’t differentiate the strata legend – can the authors use colored figure?
AU] We included a colored figure to enhance contrast between different strata.
- On Line 176 – 180 and Line 194-197 authors presented predictors of mortality, however, there is no any specific analysis plan included in the methods part. To present predictors of an outcome, the authors need to describe their approach for variable selection …what is mentioned in the methods is that they run different Cox models with d/t level of covariate adjustment. It is also not clearly mentioned which model is used to define “predictors” or which model (A, B, C) is the primary analysis? Predictor analysis requires extensive list of variables.
AU] We would like to thank the Reviewer for this comment. We included a description of the Cox Proportional Hazard model analysis in the methods section (see lines 156-163 and footnotes of table 4).
Discussion
- Can the authors discuss the high prevalence of anemia in the study cohort – 53.6% is a bit high prevalence in older adults but not uncommon. Can they add some sentences about this high prevalence?
AU] We would like to thank the Reviewer for this comment. We added a short paragraph to compare the observed prevalence of anemia with current literature (lines 275-279).
- Line 242 - include the abbreviation – “nonneuronal acetylcholine (Ach)”
AU] We corrected as requested
- Excellent discussion on the biological plausibility of AC exposure and anemia – can the authors include a summary of other mechanisms included in the discussion but not reflected in the intro part.
AU] We agree that this section was to be improved and we are grateful for suggestion. We briefly introduced potential mechanisms for biological plausibility in the Introduction in keeping with the Reviewer’s suggestion.
- Can the authors expand the clinical implication of the findings on Line 274-79 (especially focusing on strong anticholinergics with ACB 2 or more score as this was the significant one from the analysis), including reducing cumulative anticholinergic burden via deprescribing interventions in older adults with anemia as well as future research gaps (for instance further research using large sample for extended periods, confirmation study with other measures of anticholinergic burden, e.g. DBI…).
AU] We would like to thank the Reviewer for this comment. We expanded the discussion as requested (lines 331-339)
- Line 239 – spelling error ‘measures of cumulative anticholinergic burde’.
AU] We corrected as requested
Round 2
Reviewer 1 Report
The authors have done a substantial amount of revision by carefully addressing the reviewers comments. The content and clarity of the manuscript have been improved radically from the previous version.
This manuscript is a resubmission of an earlier submission. The following is a list of the peer review reports and author responses from that submission.